# The GRANDPACT Project: The Development and Evaluation of an Intergenerational Program for Grandchildren and Their Grandparents to Stimulate Physical Activity and Cognitive Function Using Co-Creation

**DOI:** 10.3390/ijerph19127150

**Published:** 2022-06-10

**Authors:** Evelien Iliano, Melanie Beeckman, Julie Latomme, Greet Cardon

**Affiliations:** Department of Movement and Sports Sciences, Ghent University, 9000 Ghent, Belgium; mebeeckm.beeckman@ugent.be (M.B.); julie.latomme@ugent.be (J.L.); greet.cardon@ugent.be (G.C.)

**Keywords:** study protocol, intervention, intergenerational, physical activity, cognitive functioning, grandparents, grandchildren

## Abstract

In recent years, increased attention has been devoted to intergenerational physical activity (PA) programs because they may have several benefits for both children and older adults (e.g., the reduction of ageism). An intergenerational PA program focusing on grandchildren and grandparents in a ‘standard’ family setting that combines PA and cognitive function is innovative and may hold potential for promoting PA and improving cognitive functioning in both grandchildren and grandparents. The aim of this study is to describe the protocol of the GRANDPACT (GRANDparents and GRANDchildren improve their Physical Activity and Cognitive functions using co-creaTion) Project, focusing on the development of an intergenerational, cognitively enriched, movement program for grandchildren and grandparents using the theoretical framework of the “Behaviour Change Wheel” in combination with a co-creation approach. Two co-creation trajectories will be organized to develop the program, followed by a pilot study to refine the program and an RCT with a pre-test (at baseline), a post-test (after 24 weeks), and a follow-up (after 36 weeks) to measure the outcomes of co-PA, cognitive functions, psychosocial well-being, and the quality of the family relationship ingrandchildren and grandparents. The outcomes will be measured using accelerometry for PA, the Cambridge Neuropsychological Test Automated Battery (CANTAB) for cognitive functions, and questionnaires for the psychological well-being and quality of the family relationship. Co-development with end-users and stakeholders during both co-creation trajectories is expected to result in an effective, attractive, and feasible program. Co-PA is expected to improve PA, cognitive functioning, psychosocial well-being, and the quality of the family relationships between grandchildren and grandparents.

## 1. Introduction

A physically active lifestyle is of great importance for the physical and psychological health of members of every age group. Some of the benefits of sufficient levels of PA in childhood include the prevention of obesity, better cardiorespiratory fitness, healthy motor development, and the reduction of symptoms of anxiety and depression [1]. According to the World Health Organization’s (WHO) physical activity (PA) guidelines launched in 2020, children should perform, on average, 60 min of moderate to vigorous PA per day and vigorous-intensity aerobic activities to strengthen bones and muscles 3 times per week. Furthermore, much attention is currently being paid to an active lifestyle as a part of healthy aging because the percentage of older adults (>65 years) continues to rise worldwide. About a fifth (20.3%) of the European Union population was 65 or older in 2019, and the percentage of people above the age of 80 is expected to double between 2019 and 2100 (from 5.8% to 14.6%) [2]. Positive outcomes of sufficient PA levels for older adults include the prevention of chronic diseases, sarcopenia, osteoporosis, social isolation, and depressive feelings [1]. Older adults are recommended to perform at least 150–300 min of moderate PA or 75–150 min of vigorous PA per week, and muscle-strengthening activities at moderate intensity 2 times per week. In addition, they should do balance, coordination, and functional exercises 3 times per week [3,4,5].

However, a lot of children and older adults do not meet these PA guidelines. No statistics are yet available on the most recent PA guidelines, but according to the guidelines from 2010, in Flanders, only 6.9% of the 6–9-year-old children, 2.6% of the 10–17-year-old children, and 12% of older adults (>75 years) met the recommended PA guidelines [6]. 

Besides the physical and psychological benefits for children and older adults, PA also has a major impact on cognitive functioning (i.e., mental process involved in gaining knowledge and comprehension, e.g., thinking, knowing, remembering, judging, and problem-solving). Examples of cognitive functions that improve due to sufficient PA include memory (i.e., storing and remembering information), executive functions (i.e., a set of mental skills, including working memory, flexible thinking, and self-control), and attention (i.e., the ability to actively process specific information in the environment while tuning out other details). It is reported that children who are more physically active perform better on attention and working memory tasks (i.e., storing and manipulating temporary information and carrying out complex cognitive tasks, e.g., reading and learning). Knowing that working memory undergoes a crucial development between the ages of 6 to 11 years and ensures better academic performance, the importance of sufficient PA during childhood cannot be underestimated [7,8,9,10,11,12]. Studies in older adults have shown that higher levels of moderate to vigorous physical activity (MVPA) and aerobics (e.g., walking, cycling, running, and sports participation) are associated with better cognitive functioning (e.g., executive functions), and, as a result, they can live independently for longer (e.g., shopping and performing household tasks) [13,14,15]. Therefore, in several systematic reviews and meta-analyses [9,16], PA has emerged as a promising approach to improving both children’s and older adults’ cognitive functioning. 

In addition to the cognitive benefits of PA, there is evidence that PA in combination with cognitively challenging activities leads to better effects in the cognitive functioning in children and older adults. In both age groups, these positive effects are explained by structural brain changes (i.e., increased brain circulation and neuroplasticity—the brain’s ability to reorganize itself by forming new neural connections throughout life) [17,18]. Older adults who are physically active in combination with cognitively challenging activities (e.g., dual tasks, exergaming, dancing, and tai chi) appear to have a better attentional aging (i.e., they can better retain their attention-paying abilities when they grow old) and improved executive functioning, which is important for their daily life performance and general health [19,20,21,22,23,24,25,26]. Yet, the effects of PA combined with cognitive activities (i.e., “cognitively enriched movement activities”) have thus far mainly been tested in controlled settings (laboratories). Moreover, only a minority of studies have examined the positive effects in children. 

In addition to the physical and cognitive benefits of sufficient PA and cognitively challenging activities described in children as well as in older adults, intergenerational programs seem to have an added value to achieving these benefits in both age groups. 

Most existing PA interventions focus on one target group (e.g., toddlers, children, adolescents, adults, or older adults). To improve communication and social interaction between different generations, an intergenerational approach has made its appearance. The idea is that two different generations (e.g., children and older adults) can perform organized activities jointly so as to create a benefit for both of them [27,28]. 

Many intergenerational studies have focused on interventions with diverse program topics (e.g., children and elderly adults cooking, reading, and learning to use a computer together). There seems to be a positive impact on children in the short- and long-term, and older adults also experienced these programs as beneficial. It has been shown that the intergenerational programs lead to children’s improved perceptions of older adults, which can lead to a reduction of ageism (i.e., stereotyping and discriminating against individuals or groups on the basis of age), and that the children learned new skills from the older adults. Older adults reported more confidence, higher levels of self-esteem and social well-being, and also improvements in memory and cognition due to helping children during an academic year [29,30,31,32]. 

Furthermore, there are a few intergenerational programs that focus on PA where benefits for both age groups have also been reported [32,33,34,35]. A tai chi program provided more PA, joy, and interaction between generations, and a dance program resulted in high levels of community engagement and enjoyment. Almost all the latter studies (except 33, 35, and 36) used a randomized controlled trial design, which is of great value to evaluating these intergenerational programs. 

Besides the scarcity of intergenerational PA programs in the literature, most of the existing literature is focused on children and older adults without a familial relationship [30]. Only one study focused on a PA program for grandchildren and grandparents in the context of kinship families, (i.e., grandparents raising their grandchildren and living with them because the parents are unable to care for them) [36]. 

In sum, while there is evidence that PA contributes to better physical, psychological, and cognitive functioning, a very high number of children and older adults do not meet the PA guidelines of the WHO. Furthermore, the combination of PA and cognitively challenging activities leads to even better cognitive functioning, and intergenerational PA programs seem beneficial for both children and older adults. Only a few intergenerational programs that focus on PA, however, are reported in the literature, and none of these programs focus on the combination between PA and cognition. Intergenerational PA programs involving familial relationships are even scarcer. Taken together, the development, evaluation, and implementation of an intergenerational, cognitively enriched PA program for grandchildren and their grandparents who are not living together is novel and could promote PA and cognitive functioning. Additionally, such a program could improve the quality of the family relationship and well-being. 

This paper outlines the study protocol for the development and evaluation of an intergenerational, cognitively enriched PA program aimed at developing an attractive and feasible program for GRANDparents and GRANDchildren to improve their co-Physical Activity and Cognition using co-creaTion (i.e., the GRANDPACT Project). To make the program more attractive and feasible, grandchildren and grandparents will be involved in the program development process to find out their needs and preferences (i.e., a co-creation approach, a detailed description of which is provided in Section 2.2.1. Co-Creation). It is suggested that the combination of PA and cognitively challenging activities will improve daily life functioning (i.e., the ability to perform certain physical and cognitive tasks faster and better) in both grandchildren and grandparents. By incorporating MVPA, coordination, balance, and strength into the program in line with the WHO PA guidelines, it is expected that a larger percentage of children and older adults will comply with the PA guidelines. Because of the intergenerational aspect set within a familial context, it is furthermore suggested that social interaction, psychosocial well-being, and attitudes towards PA in both groups will also improve, and that the program will be well-perceived by both grandchildren and grandparents. As this paper outlines a study protocol, materials and methods will be described in detail. The research intention and a discussion of it will be written out in order to consider the expected results, as well as the potential strengths and weaknesses, of the study. Results of the co-creation sessions, the pilot study, the randomized controlled trial (RCT), and the discussion of these results will be presented in other papers.

## 2. Materials and Methods

### 2.1. Materials

Overview of the GRANDPACT Project (development and evaluation)

An overview and timeline of the program development and evaluation is presented in Figure 1. First, to develop the intergenerational program, *co-creation* will be used in combination with a theoretical framework, *the Behaviour Change Wheel* [37]. Two *co-creation trajectories* will be followed: one that focuses on the development of a movement program for grandchildren and grandparents, and one that focuses on the cognitive enrichment of the latter movement program. Once both co-creation trajectories are finalized, a pilot study will be carried out to evaluate the feasibility and attractiveness of the intergenerational movement program in the target group (i.e., grandchildren and grandparents). These results will be used to further refine the program. Finally, an RCT will be conducted to measure the effects of the intergenerational movement program on the outcomes of PA, cognitive functioning, psychosocial well-being, and quality of the family relationship. The RCT will consist of two intervention groups (Arm 1: Intergenerational PA and Arm 2: Intergenerational Cognitive Enrichment) and one control group (no intervention).

Development of the intervention

#### 2.1.1. The Behavior Change Wheel

Most intergenerational programs described in the literature are neither evidence-based, nor grounded by a theoretical framework, and their effectiveness has often not been evaluated [38,39]. To develop an intervention, it is of great value to choose a useful theoretical framework. This can lead to larger effect sizes, correct responses to the right determinants, and the covering of all the important components (e.g., performing all steps of a framework in the correct order and not skipping steps) [40,41]. A theoretical framework that has frequently been used for the development of PA interventions in the past is the **Social Cognitive Theory**. However, this behavior change model was not originally designed to guide intervention development, and it has several shortcomings (e.g., the theory assumes that changes in the environment will automatically lead to changes in the person, which is not always true; the theory is based solely on the dynamic interplay of person, behavior, and environment; the theory does not focus on emotion and motivation) [42,43]. 

The **Behaviour Change Wheel** (BCW) is a theoretical framework that covers most of the important shortcomings of other behavior change models. The BCW describes eight steps that guide the development of a health intervention, from ‘Defining the problem in behavioral terms’ (Step 1) through ‘Mode of delivery’ (Step 8). An overview of the different steps is depicted in Figure 2, and the steps are discussed in detail below. First, there is a theoretical understanding of behavior to determine what needs to change; this ensures that the behavioral target is achieved. Second, a good picture is obtained about what intervention functions (e.g., education and/or increasing knowledge or understanding) are likely to be effective to bring about that change [37]. In the development of the GRANDPACT Project, these eight steps will be used to structure the co-creation trajectories.

##### The Eight Steps of the Behaviour Change Wheel

Steps 1–3: Researchers will complete Steps 1–3 of the BCW based on findings and evidence from the literature about PA, cognitive functioning, and psychosocial well-being in children and older adults.

Step 1. Define the Problem

Many children and older adults do not meet the PA guidelines of the WHO, which has negative consequences for the health of both groups (e.g., higher risk for developing obesity, diabetes, cancer, osteoporosis, and cardiovascular diseases). Besides, physically inactive individuals have a higher risk of developing cognitive decline and dementia (in older adults) and impaired academic performance (in children). Furthermore, older adults who are more socially isolated (i.e., a lack of contact with friends or family and lack of involvement in social organizations) show lower levels of PA and higher amounts of sedentary time, and report more symptoms of depression [44,45]. PA may help to reduce social isolation and feelings of depression in older adults [46], but it seems ineffective in reducing loneliness [46]. It is therefore important to focus on improving PA levels, cognitive functioning, and psychosocial well-being in both grandchildren and grandparents.

Step 2. Select target behavior

The target behavior is **co-PA**. Grandchildren and grandparents will participate in an intergenerational movement program OR an intergenerational movement program that is cognitively enriched to improve their overall PA by being physically active together. Cognitive functioning and psychosocial well-being are seen as consequences of the target behavior. 

Step 3. Specify target behavior

Grandchildren and grandparents will be encouraged to be **physically active together** in an intergenerational movement program OR in an intergenerational movement program that is cognitively enriched, of which all sessions will take place in Ghent. In addition, efforts will be made to **boost PA ‘together’ at home** (e.g., by means of a (virtual) challenge game). The frequency and duration of the sessions and the activity they will do at home will depend on the input of the target group gathered during the co-creation sessions. 

Step 4–8: Project researchers, together with participants, will complete Steps 4–8 of the BCW based on the literature about PA, cognitive functioning, and psychosocial well-being, as well as the input from the co-creation sessions.

Step 4: What needs to change?

Health behaviour is, in general, influenced by the combination of three broad categories of determinants: capability, opportunity, and motivation (COM-B). In the co-creation sessions, the researchers will investigate why people show or do not show a specific behavior based on these three types of determinants (e.g., for the health behaviour of PA, are grandchildren and grandparents able to be physically active together; do they have opportunities to be physically active together; and are they motivated to be physically active together?). The sources of behaviour will be taken into account (physical, social, psychological, reflective and automatic), but other sources of behavior will also be considered (e.g., technological), which could result in an extended version of the BCW, including new determinants. 

Furthermore, the threats, barriers, and difficulties to setting the PA behavior will be investigated using the Theoretical Domains Framework (TDF). These TDFs are subcategories of the three broad categories. Examples of TDFs are given in Figure 3 [47]. 

The primary outcomes will be PA and cognitive functioning in both grandchildren and grandparents; the secondary outcomes will be psychosocial well-being and the quality of the family relationship between grandchildren and grandparents. We expect to see improved levels of PA, cognitive functioning, psychosocial well-being, and quality of the family relationship in the intervention group. Cognitive functioning will improve more in grandchildren and grandparents who participate in the cognitively enriched intergenerational movement program. 

Step 5: Intervention functions

The project researchers will consider the cases that emerged from Step 4 and will determine which intervention function is best suited to change the behaviour (i.e., a lack of PA). Intervention goals will also be drafted, which will then be presented to the grandchildren and grandparents in the co-creation sessions, where they can give their input and feedback. 

Step 6: Policy categories

In this step, policy categories will be identified. The policy categories will need to support the intervention functions in Step 5. It will be of great value that the sessions will be embedded in an organization afterwards so as to ensure sustainable implementation. Some examples of organizations and policy categories are Family Sport Flanders, senior and youth sports, and municipal sports.

Step 7: Behaviour change techniques

The project researchers will aim to identify the most common behaviour change techniques (BCTs) (e.g., self-reward, conditioning, punishment, and focus on past success) from the list of 93 possible BCTs [48] based on the intervention goals that have been established in previous steps. Michie et al.’s (2013) taxonomy of BCTs will be used to select the appropriate BCTs. This taxonomy describes how each technique is linked to different intervention functions. 

Step 8: Mode of delivery

After identifying the BCTs, it will be decided how those BCTs will be turned into practical strategies.

#### 2.1.2. Co-Creation

A lot of interventions are developed using a deficit-based approach, where researchers use their knowledge to implement a general program for the participants. However, it has been shown that developing an intervention without taking the needs and requirements of the target population into account results in higher drop-out and lower participation rates for the program [49,50]. A design process in which end-users are engaged in the development process is called a co-creation design. It is a co-construction between researchers, end-users (i.e., grandchildren and grandparents), and stakeholders (i.e., experts in movement sciences and organizations) with the aim of tailoring the program to the target group. Co-creation has received more attention in recent years to enhance effectiveness and to prevent drop-out in (health promotion) interventions [51,52,53,54]. A co-creation design will be specifically used to empower grandchildren and grandparents, to learn what kind of activities they would like to do in the program, to learn what their ideal program looks like, and to keep fun levels and motivation and participation rates high in order to make the intervention effective and to obtain implementation success. More detailed descriptions of the co-creation trajectories and sessions are presented below. 

As mentioned earlier, two co-creation trajectories will be followed. The first one will focus on the development of an intergenerational movement program (intergenerational PA); the second one will focus on the cognitive enrichment of the intergenerational movement program (intergenerational cognitive enrichment). Co-creation Trajectory 1 will include a series of 5 sessions, each lasting an average of 90 min, at intervals of 2 weeks so as to provide enough time to prepare for each subsequent session. As we are working with children, the sessions must be especially creative and visual, which means time will be needed to craft (e.g., cutting, drawing, and painting) between the sessions [53]. The first and third sessions will involve grandparents only; the second and fourth sessions will involve grandchildren only, and the last will be held with both. (The format of Co-creation Trajectory 2 will depend on the output of Co-creation Trajectory 1 and will probably consist of only 3 sessions). The purpose of these sessions will be to identify the needs and requirements of the grandchildren and grandparents as they relate to PA, cognitive functioning, the intergenerational aspect, and the content of the program. Grandchildren and grandparents will be separated in the sessions in order to create a safe environment where they can express their opinions. They will be updated about each other by the researcher in the beginning of each session. Researchers, who will have received training in co-creation (e.g., a webinar about participatory research), will lead the sessions to engage and motivate the participants to share their opinions and cooperate.

#### 2.1.3. TIDieR Checklist

To allow intervention replicability, the Template for Intervention Description and Replication (TIDieR) will be used to describe our intervention and to indicate the location of information [55] (See Appendix A).

### 2.2. Methods

#### 2.2.1. Co-Creation

##### Participants and Sample Size

Similar inclusion criteria will be applied for participation in both trajectories (intergenerational PA and intergenerational cognitive enrichment). Grandchildren will be 6 to 10 years old; grandparents will probably be aged between 60 and 75, but their age will depend on the age of the children. This age range for the grandchildren has been chosen because, in Flanders, the majority of children in this age category fail to reach the guidelines for PA; only 6.5% of the 6–9-year-old children have an adequate PA level [56]. Furthermore, interventions to promote PA in this age group can be improved to sustain longer-term effects [57], and young children are still influenceable, which makes it advantageous to include them in this intervention. In addition, it is a good age range for choosing appropriate work formats for the co-creation sessions, and grandparents of children in this age group are normally still physically able to participate in a movement program. To explore what grandmothers and grandfathers like, both genders will be recruited in the co-creation sessions. Inclusion criteria for the grandchildren include: an age between 6 and 10 years old; the possession of at least 1 grandparent; the ability to speak Dutch; and the lack of any serious physical, cognitive, or psychological health problems. Inclusion criteria for the grandparents include: the possession of at least 1 grandchild (6–10 years old), the ability to speak Dutch; and the lack of any serious physical, cognitive, or psychological health problems. 

Each co-creation trajectory will include 12 grandchildren and 12 grandparents, according to Leask’s recommendations for sample sizes in co-creation activities [58]. (See Figure 4).

Study Design and Numbers of Participants

##### Recruitment of Participants

Convenience sampling will be used to recruit participants for both co-creation trajectories. A first strategy will be to reach out to children via schools in the city of Ghent and to contact their grandparents (via their parents). All of the children of the first and the second grades (classes 1–4 of primary school) of 2 schools will receive an information letter about the goal of the study and some explanation about the co-creation trajectories. In the case that this results in too few respondents, additional schools will be contacted. If both children and their grandparents are interested in participating (parents must ask the grandparents), they can fill in their contact information using a paper strip attached to the information letter. Of those who respond with interest in joining, a researcher will call the parents and grandparents to ask some questions regarding the sociodemographic characteristics (i.e., age and gender) and inclusion criteria (i.e., language and physical, cognitive, and psychological health status) of the grandchildren and grandparents. A group of 12 grandchildren and 12 grandparents will be chosen by purposive sampling, which means that a researcher will choose the participants based on these characteristics/criteria so as to better represent the population during the co-creation trajectories. In regard to age, both younger and older grandchildren and grandparents will be chosen. In regard to gender, there will be a mix of boys and girls and grandmothers and grandfathers.

Before participating in the co-creation trajectories, each participant will be asked to sign an informed consent. Grandchildren will need parental consent, as well as their own informed consent; grandparents will sign an informed consent for themselves. Grandchildren and grandparents will be asked to bring these signed forms to the first co-creation session that they attend.

##### Content of the Co-Creation Trajectories


Co-creation Trajectory 1: Development of the Intergenerational movement program.


*Session 1 (grandparents):* In this session, researchers and grandparents will meet each other. A short introduction about the intergenerational movement program will be given. Grandparents will be asked to write down movement activities that they would like to do (and not like to do) with their grandchildren. The barriers and motivators will be determined, which will be the implementation of Step 4 of the theoretical framework. COM-B (i.e., capability, opportunity and motivation of behavior) and the TDF (i.e., Theoretical Domains Framework) will be examined in order to obtain a better overview of their ability and potential (e.g., ‘Do you feel you have enough skills to exercise together with your grandchild? Why or why not?’) and the barriers and facilitators (e.g., ‘Does the environment offer sufficient opportunities to be active together? Why or why not?; ‘Do you have sufficient time to exercise together? Why or why not?’; ‘Are you able to sum up some advantages to being physically active together with your grandchild?’). 

*Session 2 (grandchildren):* In this session, the same structure will be used as that used in the first session with grandparents. The biggest difference between the two sessions will be that, in this session, the information will be obtained by playing games and making metaphors to motivate the children during the sessions, and a mascot—a kangaroo with a child in its pouch—will be used. A kangaroo was chosen because they are quite active animals, and it will represent the grandparent with the grandchild.

After these two sessions, the intervention goals will be established/refined by the project researchers based on the barriers and motivators expressed by the grandchildren and grandparents. Furthermore, the intervention functions will be considered and selected based on Step 4 of the BCW. Both intervention goals and functions will be discussed with the grandparents; only intervention goals will be discussed with the grandchildren in the following co-creation sessions. 

*Session 3 (grandparents):* In this session, first a summary of Sessions 1 and 2 will be given. Grandparents will be able to give their input and feedback on the proposed intervention goals and intervention functions, which is Step 5 of the BCW. Second, the ideal movement session will be explored. More specifically, **content** and **format** (e.g., working on a theme, the specific topic per session, game format or exercises, everything being done together or being split between grandchildren and grandparents); **frequency** and **duration** (e.g., how many hours and how many times per week); **intensity** and **environment** (e.g., cardio, balance, coordination, a combination of both, high sweat production, indoor or outdoor, in a green or urban area) will be examined. Finally, we will explore which movement activities the grandchildren and grandparents can do together at home in addition to the sessions. Researchers will give some examples, and grandparents will be asked to give their opinions (e.g., pedometer, treasure hunt, challenge: goalsetting for taking more steps/day, games, using an app for workouts, going online to do exercises together via Zoom). In addition, grandparents will be asked how they can be **motivated** to participate in the sessions to keep participation rates high. 

*Session 4 (grandchildren):* In this session, the same structure will be used as in the third session with the grandparents. As in the previous session with the grandchildren, information will be obtained in a more informal way, using games, photos, and interactions with each other. Some of the elements that were explored in the session with the grandparents (e.g., intervention functions, frequency, and duration of the sessions) will not be discussed with the grandchildren because that would be too difficult for them. 

After these 4 sessions, a movement session will be prepared by the project researchers and movement scientists based on the input and feedback from Sessions 1–4.

*Session 5 (both groups):* In the last session, grandchildren and grandparents will be brought together. A movement session characteristic of the program will be presented to them. Some exercises will be tried out, and grandchildren and grandparents will be asked a few questions about attractiveness, feasibility, fun, motivation, and co-PA. At the end, they will fill in a SWOT analysis to determine the strengths, weaknesses, opportunities and threats that they associated with the exercises. 

*Co-creation Trajectory 2: Cognitive enrichment of the intergenerational movement program*. This co-creation trajectory will follow the same order and procedure as described for the first co-creation trajectory, but the information that was already obtained will be included here, and it will not be collected again. Therefore, it is expected that only three sessions will be needed in order to develop the cognitively enriched intergenerational movement program. 

*Session 1 (grandparents):* In this session, the activities that were performed and the most important findings from the previous co-creation trajectory will be explained. Furthermore, the objectives of this trajectory and the expectations will be clarified. The main goal is to find out which cognitive movement activities they found to be attractive and feasible for inclusion in this program. An introduction to PA and cognitive functioning will be given, and researchers will also provide some examples of cognitively enriched movement activities. The desired duration and challenge of the activities will be ascertained. From the literature, it is known that a combination of PA and cognitive functioning must last at least 10 minutes to observe effects, so this will be taken into account. 

*Session 2 (grandchildren):* This session will have the same content as the session with the grandparents. A big difference will be that the desired duration and challenge will not be ascertained from the children. Furthermore, everything will be explained in an understandable way for them. More examples of cognitively enriched movement activities will be provided to the children as developed by both the researchers and the grandparents (from Session 1). 

*Session 3 (both groups):* As in the previous co-creation trajectory, grandchildren and grandparents will be brought together, and the cognitively enriched movement activities for the intergenerational movement program will be tested out. Grandchildren and grandparents will be asked a few questions about attractiveness, feasibility, fun, motivation, and co-PA. At the end, they will be asked to fill in a SWOT analysis to determine the strengths, weaknesses, opportunities and threats that they associated with the exercises. 

##### Process Evaluation Co-Creation

After each co-creation session, participants will be asked to fill in a short questionnaire to evaluate how the session was perceived (i.e., ‘How do you feel at this moment?’; ‘Would you give/not give your opinion during the session, and why?’; ‘Did the environment feel safe for you, and why?’; ‘During which portion, or with which aspect, did you feel uncomfortable during the session?’; ‘Which aspects of the session were fun for you, and which aspects were boring?’; ‘Which aspects caused you to be motivated to do something?’; ‘Why did you take part (or not take part) in conversations?’). They will also be able to add suggestions for the next sessions (i.e., ‘Can you add some suggestions for the next session(s)? What could be improved? What should be different?’). 

##### Expert Meeting 

At the end of Co-creation Trajectories 1 and 2, an elaborated series of activities will be prepared by movement scientists, after which an expert meeting will be organized with experts in movement activities and cognition in children and older adults in order to obtain additional advice about the proposed activities (e.g., ‘What is important in some specific exercises?; What is the correct body posture?; Which elements are important for motivation or persistence in something?’). An observer will write down the most important findings. 

Evaluation of the intervention

#### 2.2.2. Pilot Study

After the two co-creation trajectories are finalized, a pilot study will be conducted to determine whether the two different programs are attractive, feasible, and fun for grandchildren and grandparents (see Figure 1). The purpose of this portion of the study will be to finetune the programs and recruitment strategies to optimize the RCT. There will be 2 groups of 12 dyads. Group 1 will engage in the intergenerational PA program, while Group 2 will engage in the cognitively enriched intergenerational PA program. Each session will be led by researchers and facilitators (with a background in movement and sports sciences and/or health promotion research). All the sessions will take place in Ghent (i.e., where the main researchers are based); the exact location will depend on the preferences of the participants and what is practically feasible.

##### Participants and Sample Size

Similar inclusion criteria will be used to recruit participants for the pilot study as those used for the co-creation trajectories. (See Participants and Sample Size Part in Section 2.2.1) The required sample size for a pilot study is between 30 and 50 participants [59]. Therefore, and also taking into account a group size where safety can be guaranteed, two groups of 12 dyads will be recruited (i.e., in total, about 24 grandchildren and 24 grandparents; 12 grandchildren and 12 grandparents per group, including the participants of the co-creation sessions), assuming an attrition rate of 30% (7 dyads) [61], and guaranteeing a sufficiently large group size for the sessions. A projected attrition rate of 30% instead of 20% has been chosen because we have taken into account that, if one grandchild or grandparent drops out, this implies that the whole dyad will be excluded from the study. Therefore, 2 groups of 12 dyads will be recruited (a total of about 24 grandchildren and 24 grandparents). (See Figure 4).

##### Recruitment of Participants 

Similar recruitment strategies will be used to recruit participants for the pilot study as those used for the co-creation trajectories. (See Recruitment of Participants Part in Section 2.2.1) If the recruitment strategy of solely using schools is insufficient to obtain the required number of participants, an additional strategy will be used: launching a call for grandparents via Ghent University Research Consortium for Aging Young (GRAY) and its partners (e.g., the Flemish Council on Aging). GRAY is an interdisciplinary research consortium on healthy aging. Eligible grandparents will receive an information letter and fill in contact information for their grandchildren and themselves (both must be able to participate) to be returned to the researchers. A researcher will call the grandchildren’s parents and grandparents to probe for sociodemographic characteristics and inclusion criteria. There are no specific restrictions regarding age or gender in the pilot study, which differs from the co-creation sessions, where limits are placed on the numbers of younger and older children and there will be a good mix between boys and girls and grandmothers and grandfathers. 

##### Content of the Sessions

A total of about six sessions per group (e.g., one session per week) will take place over the course of six weeks (depending on input from the co-creation sessions) at a preferred time during the week. Group 1 will engage in the intergenerational movement program; Group 2 will engage in the cognitively enriched intergenerational movement program. The duration and content of the sessions will depend on the needs and preferences of the target group and the intervention goals that were set during the co-creation sessions (e.g., movement activities: balancing on a bench, relays, ballgames; and cognitively enriched movement activities; walking/running games, searching for objects in the environment, coordinative exercises while counting). According to the recent PA guidelines of the WHO for children and older adults, researchers will integrate coordination, balance, mobility, and strength into the sessions.

##### Measurements

Process Evaluation

After each session, a short questionnaire will be administered to probe for the thoughts and feelings of the grandchildren and grandparents regarding attractiveness and feasibility during the activities and to determine which kinds of topics or activities they prefer the most (e.g., games, exercises, activities with one’s own grandchild or also with the other grandchildren). The Physical Activity Enjoyment Scale (PACES) will be used to assess the level of enjoyment among grandparents. This short, validated questionnaire for older adults includes eight items that will assess how much fun the grandparents experienced during the (cognitively enriched) movement activities [62]. The use of emoji questionnaires with younger children can measure their emotional responses and can also help them to understand abstract concepts and express themselves. Therefore, smileys will be used to express the level of enjoyment in children [63]. After the six sessions, an additional, self-administered questionnaire will evaluate participation in the sessions, recruitment strategy, group size, age, and the clarity of the questionnaires that have been administered after the sessions. A focus group with at least three stakeholders (e.g., MOEV, Gezinssport Vlaanderen, and Sportwerk Vlaanderen) will be organized to share ideas and to receive expertise to finetune the activities of the preliminary movement program. An observer will write down the most interesting points. MOEV is an abbreviation for Motivatie (Motivation), Ondersteuning (Support), Expertise (Expertise) and Vernieuwing—Vlaanderen—Vitaliteit (Renewal—Flanders—Vitality). MOEV is an organization that motivates and supports schools in the development of their exercise and sports policies; the aim is that toddlers, primary school children, and adolescents will have fun by being physically active.

Evaluating the use of outcome measures for the RCT

To objectively measure PA in the grandchildren and grandparents, the Axivity AX3 accelerometers or the ActiGraph wGT3X-BT will be used. Half of the group doing the intergenerational movement program will wear the first accelerometer; the other half will wear the second one. During the cognitively enriched sessions, half of the group will also wear the first one; the others will wear the second one. The most feasible, practical, and useable of the accelerometers will be chosen to be used as a measure in the RCT. These accelerometers will be used in combination with a seven-day diary (pre- and post-follow-up) to subjectively measure PA, and they will be worn during certain sessions. More details about the measurement instruments are described below in Outcome Measures part of Section 2.2.3.

To objectively measure cognitive functioning in the grandchildren and grandparents, CANTAB tests will be used. The purpose will be to assess the feasibility, usability, and practicality of the test battery. The pilot study will be used to make a final selection of which cognitive tests to include in the RCT and decide on the ideal total duration of the cognitive testing procedure (e.g., +/− 40 min) based on practicality and discriminating power. More details are described in Outcome Measures part of Section 2.2.3.

To measure psychosocial well-being in the grandchildren and grandparents, several questionnaires will be administered. The preferred mode of delivery (online or paper-and-pencil questionnaires), the number of questionnaires, and the total duration of completing the questionnaires will be evaluated. Potential unclarities or difficulties when completing the questionnaires will be detected in order to improve this in the actual RCT. More details are described below in Outcome Measures part of Section 2.2.3.

Furthermore, participants will be asked for input and feedback on the measures and measurement instruments using a short questionnaire.

##### Achieving the Intervention Goals

Depending on the findings that emerge from the co-creation sessions, a gadget or material (e.g., stickers, magnets, eyecatchers, or posters) can be developed to optimally achieve the intervention goals. A self-administered questionnaire will assess whether these program materials are perceived as positive and supportive to achieving the intervention goals.

##### Data Analysis

The answers on the open-ended questions in the questionnaires will be coded using content analysis. The answers on the Likert scale questions in the questionnaires will be coded using SPSS [64]. Multilevel linear analysis will be used to measure the effects of co-PA.

#### 2.2.3. Randomized Controlled Trial (RCT)

The RCT will be conducted subsequent to the pilot study so as to measure the effects of the two different programs on the outcomes of PA, cognitive functioning, psychosocial well-being, and quality of the family relationship in the grandchildren and grandparents and to evaluate the process of the study. The RCT will consist of two intervention groups, Arm 1: Intergenerational PA and Arm 2: Cognitively enriched intergenerational PA, and one control group. Intervention Group 1 will receive the intergenerational movement program, Intervention Group 2 will receive the cognitively enriched intergenerational movement program, and the control group will receive no intervention. The pre- (before the start of the sessions), post- (after 24 weeks) and follow-up (after 36 weeks) outcome measures of PA, cognitive functioning, psychosocial well-being, and quality of the family relationship will be compared between Group 1 and the control group, between Group 2 and the control group, and between Groups 1 and 2. An overview of the RCT is shown in Figure 5.

##### Participants and Sample Size

The same inclusion criteria will be used for the RCT as in the co-creation trajectories and the pilot study (i.e., speaking Dutch; having at least one grandparent or grandchild; and having no serious physical, cognitive, or psychological health problems). Based on effect sizes of intergenerational programs of children and older adults, an effect size of 0.15 will be intended for the outcomes of PA, cognitive functioning, psychosocial well-being, and quality of the family relationship [65]. The required sample size has been calculated using the software GPower 3.1.9.7 (Universität Düsseldorf, Düsseldorf, Germany) [60]. The sample calculation will be based on 80% power to detect a significant difference in PA, cognitive functioning, psychosocial well-being, and quality of the family relationship between groups and over time. Significance level alpha 0.05, having 3 groups (2 intervention groups and 1 control group) and 3 measurements (pre-, post-, and follow-up). A priori power analysis for the RCT suggests a total sample size of around 45 dyads (*n* = 90 grandchildren and grandparents). Assuming an attrition rate of 20% [61], a total sample size of 54 dyads (*n* = 108 grandchildren and grandparents) will be required for the 3 groups. (Each group consists of 36 participants: 18 grandchildren and 18 grandparents). See Figure 4.

##### Recruitment of Participants

According to the evaluation of the recruitment strategy of the co-creation sessions and the pilot study, the best way to recruit the target population will be chosen based on convenience sampling. The strategies and procedures for the recruitment are described above in the discussion of the recruitment for the co-creation trajectories and pilot study. The first strategy will be the recruitment of grandchildren and their grandparents via schools in (the neighborhood of) Ghent. The second strategy will be via GRAY and their partners. A combination of both strategies is also possible if needed to achieve the desired number of participants.

##### Content of the Sessions

A total of 12 (or 24) sessions (1–2 session(s) every (1 –2) week(s) per group will take place over 24 weeks at a preferred time during the week (e.g., Wednesday afternoon, Tuesday evening, Thursday evening, etc.). All of this will depend on the preferences of the grandchildren and grandparents. Arm 1 and Arm 2 will receive different sessions during the same week. Arm 1 will receive only movement activities, while Arm 2 will receive cognitively enriched movement activities. The content of the sessions will have been refined during and after the pilot study to best meet the needs and preferences of the grandchildren and grandparents (e.g., duration, topics, and format). After the follow-up measurements, the intervention groups will also receive a brochure detailing (cognitively enriched) movement activities that they can perform at home to achieve a long-term effect.

##### Measurements

Process evaluation

Similar to the process evaluation in the pilot study, attractiveness, feasibility, and enjoyment will be examined after each session of the finalized intergenerational movement program.

Effect evaluation

PA, cognitive functioning, psychosocial well-being, and quality of the family relationship will be measured in the grandchildren and grandparents to determine possible effects. A more detailed description is given in Outcome Measures Part in Section 2.2.3.

Outcome measures

According to the results of the accelerometers used in the pilot study, **physical activity** will be objectively measured in the grandchildren and grandparents using Axivity AX3 accelerometers or ActiGraph wGT3X-BT. These validated accelerometers in children and older adults [66,67,68] must be worn pre- (before the start of the sessions, at baseline), post- (right after the RCT ends, after 24 weeks), and for a third period at follow-up (after 36 weeks) for 7 consecutive days. This will be performed so as to examine changes in physical activity levels in the grandchildren and grandparents (due to the movement sessions and the encouragement to be more physically active at home) and to compare the effects of the intervention with the control group. The locations on the body will be on the wrist of the non-dominant hand for AX3 accelerometers [69,70] and on the right hip for the ActiGraph wGT3X-BT accelerometers [71]. The Bluetooth function of the ActiGraphs can give information about co-PA between the grandchildren and grandparents, but the feasibility of using this function in practice will be assessed. Additionally, a seven-day diary will be used to study the context of co-PA. For all activities, grandparents do (virtually) together with their grandchildren, they will write down which activities they performed, the start and end times of their activities, and their ratings of the perceived intensity of these activities. To monitor the intensity of PA of activities that the grandchildren and grandparents do together, the grandchildren and grandparents will also wear the accelerometers during certain sessions.

**The Cognitive functioning** of the grandchildren and grandparents will be objectively quantified pre- (before the start of the sessions, at baseline), post- (after 24 weeks), and at follow-up (after 36 weeks) by means of a validated neuropsychological test battery (i.e., CANTAB). The CANTAB is sensitive to changes in neuropsychological performance and allows the differentiation between different cognitive functions. The standardized and automated CANTAB can be used in children from the age of 4 and has been used and recommended in research in older adults [72,73]. Several specific cognitive functions, including 4 domains (i.e., ‘practice’, ‘attention’, ‘memory’, and ‘executive function and decision making’) will be objectively assessed using an iPad 10.2 with the validated CANTAB software (cambridgecognition.com, accecced on 12 January 2022).

**Psychosocial well-being** will be measured by several questionnaires that measure well-being indicators relevant to children and older adults. For children, the 23-item PedsQL (Pediatric Health-Related Quality of Life) scale will be used to obtain a reliable and valid global assessment of the children’s general quality of life and their specific emotional, social, and school functioning [54]. For older adults, general psychological well-being, depression, stress, anxiety, and social isolation will be measured. First, to capture a wide conception of older adults’ psychological well-being, including affective–emotional aspects, cognitive–evaluative dimensions, and psychological functioning, the Warwick–Edinburgh Mental Well-Being Scale (WEMWBS) will be used [51]. Symptoms of depression, (dis)stress, and anxiety will be measured using the Dutch version of the valid and reliable Four-Dimensional Symptom Questionnaire (4DSQ) [52]. Social isolation (defined as the way in which older adults perceive, experience, and evaluate isolation and lack of communication with others) will be measured using the valid and reliable Social and Emotional Loneliness Scale for Adults (SELSA) [53].

**Quality of the family relationship** will be measured in children with the Grandparent–grandchild Relationship Questionnaire using 3 sections (sociodemographic information, family, and participation in certain activities) and Rey and Ruiz’s Socialization Styles Questionnaire, which consists of 34 items on a 4-point Likert scale, and which evaluates democratic and authoritarian socialization styles [74]. In grandparents, this will be measured with a self-report questionnaire based on the previous investigations of Farneti and Battistelli. This 26-item questionnaire will assess the grandparents’ opinions concerning their relationships with their grandchildren using 3 subscales [75].

##### Data Analysis

Multilevel linear analysis in R will be used to estimate the between and within subject effects on PA, cognitive functioning, psychosocial well-being, and the quality of the familial relationship in grandchildren and grandparents. The mediating role of co-PA in the relationship between PA, cognitive functioning, psychosocial well-being, and the quality of the family relationship will be statistically checked in R using structural equation modelling (path analysis). Open-ended questions will be conducted using content analysis in SPSS (i.e., creating a code tree and looking for themes in the responses) [64].

#### 2.2.4. Data Management

All data will be stored on a password-protected computer and a central hard disk of the research group. The data of the participants will be pseudonymized, and only the main researcher will have access to the key to link raw data on personal data. Data will be stored for a minimum of 5 years after the appearance of publications based on these data. After this period, the data can be destroyed.

#### 2.2.5. Ethics Approval and Consent to Participate

This study and its three parts (co-creation, pilot study, and RCT) have been submitted for approval by the Committee of Medical Ethics of the Ghent University Hospital. Ethical approval has already been received so as to coincide with the beginning of the co-creation sessions (EC number: BC-11748). The RCT will be registered at ClinicalTrials.gov, which is a database of privately and publicly funded clinical studies conducted around the world. Participants in the co-creation sessions, the pilot study, and the RCT will receive an information letter and informed consent to provide them with all of the details about the study. Grandparents and parents will be asked to read and sign the informed consent; grandchildren will be asked to sign an informed assent before the start of the co-creation sessions, pilot study, and RCT. Each participant will be informed about the purpose of the study, the design, the procedure, the right to leave the study at any time, and the data confidentiality.

## 3. Research Intention

The research intention of the program development and evaluation is a theory-based lifestyle program which aims to improve Physical Activity and Cognitive functioning in GRANDchildren and their GRANDparents using co-creaTion (i.e., the GRANDPACT Project).

For the program development, it is expected that a theoretical framework, in combination with a co-creation approach, will lead to an attractive and feasible intergenerational movement program in which grandchildren can participate together with their grandparents. The needs and requirements of end-users will be taken into account, and this will make them feel empowered. For the program’s evaluation, it is expected that the pilot study will help to refine the movement program that will be drafted after the co-creation sessions. Furthermore, the refinement of measurement instruments, recruitment strategies, ages, and group sizes will be important to optimize the program for the RCT. The RCT will be conducted to evaluate effects of the (cognitively enriched) intergenerational movement program on co-PA, cognitive functioning, psychosocial well-being, and quality of the family relationship.

Arm 1 of the RCT will mainly focus on increasing co-PA (i.e., PA between grandchild and grandparent), while Arm 2 of the RCT will mainly focus on increasing cognitive functioning in both the grandchildren and grandparents via a *cognitive enrichment* of co-PA (i.e., cognitively enriched PA between grandchild and grandparent). The main goal and primary outcome of the intervention is to improve co-PA in grandchildren and grandparents (based on the WHO PA guidelines), and taking feasibility into account, we aim to reach a level where grandchildren are physically active together with their grandparents 2 times per week, for a minimum of 30 min of performing MVPA. The secondary goal is to improve total PA, cognitive functioning, psychosocial well-being, and quality of the family relationship in grandchildren and their grandparents, and it is expected that co-PA (over a period of 36 weeks) will act as a mediator in this relationship. It is hypothesized that Arm 1 (focusing on intergenerational PA) and Arm 2 of the RCT (focusing on intergenerational cognitively enriched PA) will lead to a greater improvement in PA, cognitive functioning, psychosocial well-being, and quality of the family relationship than that of the control group (no intervention). Furthermore, it is expected that the cognitive enrichment of the intergenerational movement program will lead to a greater improvement in cognitive functioning in Arm 2 of the RCT.

## 4. Discussion of the Research Intention

This paper describes the study protocol of the GRANDPACT Project. The main aim of this project is to develop an effective, attractive, and feasible (cognitively enriched) intergenerational movement program for grandchildren and their grandparents to improve their co-PA and cognitive functioning (primary outcomes), psychosocial well-being, and the quality of the family relationship between the grandchildren and their grandparents (secondary outcomes).

### 4.1. Importance of This Study

The development of this intergenerational movement program for grandchildren and grandparents is innovative for the following four reasons. First, to our knowledge, this will be the first intervention that uses the Behaviour Change Wheel in combination with a co-creation approach to develop a cognitively enriched intergenerational movement program for grandchildren and grandparents. Previous intergenerational interventions described in literature are often not evidence-based or grounded by a theoretical framework, and if so, mostly the Social Cognitive Theory was used for the development of the PA intervention [42,43]. However, this behaviour change model has some shortcomings, which are covered by the Behaviour Change Wheel [37]. The Run Daddy Run intervention used the Behaviour Change Wheel in combination with a co-creation approach to develop an intervention targeting PA in fathers and children (co-PA) to improve children’s lifestyle behaviours. This combination seemed to be effective in developing this intervention and reaching the expected goal [76]. Therefore, and also because this study focuses on family members, this method has been chosen for this study. Second, to our knowledge, grandchildren and grandparents who do not live together have never been involved in the development of an intergenerational movement program. In the literature, only one program focused on intergenerational PA, but in the context of kinship families, where grandparents are raising their grandchildren and living with them [36]. However, this situation is not a ‘standard situation’ in Belgium, nor is it common in other, large parts of the world where most grandchildren live apart from their grandparents. The main finding of this study, which focused on grandparents raising their grandchildren, is that the community-based participatory research (CBPR) framework seems to be an effective approach to improving health outcomes and to implementing the intervention, which reaffirms the value of the co-creative approach in developing health promotion interventions. Furthermore, it is expected that intrafamilial relations may strengthen the motivation to be physically active together and to make a long-term commitment. Thirdly, the quality of the family relationship between grandchildren and grandparents, which will be investigated in this study, has already been explored [74,75], but not in a context in which they engage in organized activities together. It can provide an interesting insight into how the relationship between them changes from the way it was before they performed the movement activities together. Lastly, the combination of PA and cognitively enriched activities has never been used in intergenerational research. The focus in intergenerational programs mainly consists of doing organized activities together. Most of the time, these are creative, physically active, or cognitively stimulating activities that children and older adults perform together [65]. Both PA intergenerational programs and intergenerational programs focusing on cognitive functioning alone do exist, but none of these programs uses the combination of PA and cognitive functioning in the same program, while there is evidence that the combination of PA and cognitively enriched tasks results in even better cognitive functioning [17,21]. Therefore, the use of this combination can lead to a greater improvement in the cognitive functioning of both the grandchildren and the grandparents.

### 4.2. Potential Strengths and Limitations of This Study

A first potential strength of this study includes the experimental and longitudinal design. A second potential strength is the use of the Behaviour Change Wheel in combination with a co-creation approach in which end-users and stakeholders will collaborate in the development of the movement program. This may lead to a more effective intervention, and it will also prevent drop-out because the needs and preferences of the grandchildren and grandparents will be taken into account [51,52,53,54]. In addition, organizations working with children and older adults (stakeholders) will be asked to offer information about the usefulness, attractiveness, and feasibility of the movement activities (i.e., a bottom-up instead of a top-down approach, which means that researchers base the research upon the end-users, rather than upon a general theory or idea based on findings in the literature) [77]. A last potential strength is that PA and cognitive functioning will be measured objectively through accelerometry and CANTAB tests, while self-report measures (e.g., questionnaires and diaries) can lead to response and recall biases [78].

A first potential limitation of this study includes the lack of blinding, as the researcher will be involved in the data collection and analysis. Therefore, a strict protocol has been created so as to reduce this limitation as much as possible. A second potential limitation includes the recruitment strategy. Convenience sampling will be used for the co-creation sessions, the pilot study, and the RCT. It is therefore expected that grandchildren and grandparents who are already more physically active and motivated to move will be interested participating. However, this will be taken into account during the recruitment by mentioning that the levels will be attainable for everyone and that the focus will not only be on the movement activities, but also on creating a pleasant atmosphere by making the activities fun, attractive, and feasible for the target group.

### 4.3. Implementation of This Intervention

Implementation of the intervention program is likely to impact its success; therefore, several strategies will be used to facilitate correct implementation of the intergenerational movement program by different organizations (e.g., Family Sports Flanders and Sports Flanders). First, a detailed protocol will be written to assure that the intervention will stay theory-based and to minimize attrition rates, which will both foster the success of the intervention (i.e., this protocol will be the common thread; researchers, stakeholders, grandchildren, and grandparents should respect this protocol). Second, the expert meetings with diverse stakeholders from different organizations (e.g., GRAY, Family Sports Flanders, and MOEV) that will be held after the co-creation trajectories will help to gain feedback and additional advice about the feasibility of implementing the cognitively enriched movement activities for grandchildren and grandparents (e.g., body posture, execution of movements, and other aspects that require attention). This input will also be integrated into the protocol. Third, the planned pilot study will help to refine the cognitively enriched movement activities to ensure that they will be useful and correctly executed, as well as attractive and feasible for the target population. If results of the RCT show that the intervention is indeed effective at improving co-PA and cognitive functioning in the first place, and psychosocial well-being and the quality of the family relationship in the second place, a meeting with organizations who are interested in implementing this intergenerational movement program will be held to discuss the information necessary for grandparents, parents, and children, which materials are needed, which locations are preferable and feasible, which certified teachers should be recruited, etc. A very important factor is that the embedding of this project in GRAY will help to engage Flemish partners, such as Family Sports Flanders and Sports Flanders, in its execution, dissemination, and valorization.

## 5. Conclusions

This will be the first intergenerational movement program targeting grandchildren and grandparents using the BCW in combination with a co-creation approach and using a combination of PA and cognitively enriched PA. Developing and evaluating this intergenerational, cognitively enriched movement program might be effective in improving co-PA, cognitive functioning, psychosocial well-being, and the quality of the family relationship in both grandchildren and their grandparents. If the program seems to be effective, it could be implemented in different organizations in Flanders, and it could have important implications for future research and health policies. Targeting grandchildren and their grandparents for participation in an intergenerational cognitively enriched movement program might be a novel and effective approach to improving general health in both age groups.

## Figures and Tables

**Figure 1 ijerph-19-07150-f001:**
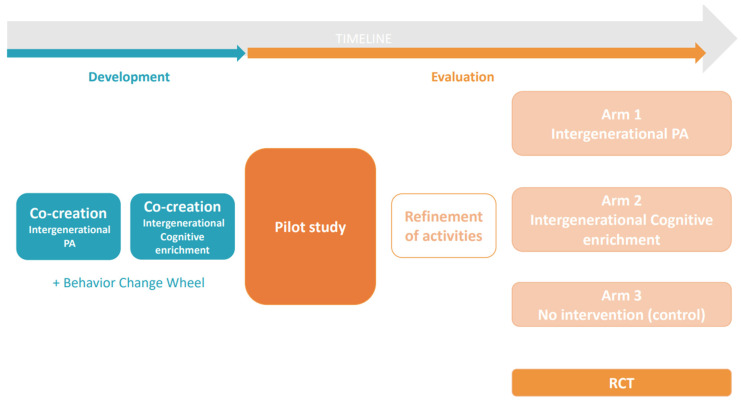
Overview and timeline of the GRANDPACT Project.

**Figure 2 ijerph-19-07150-f002:**
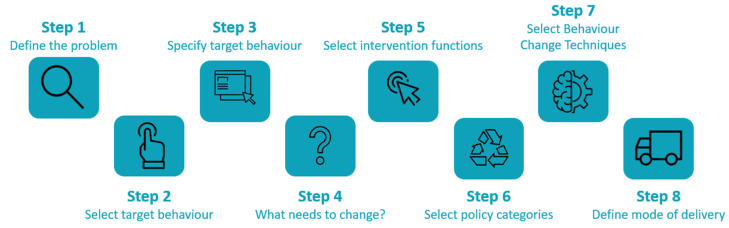
The eight steps of the Behaviour Change Wheel used in the GRANDPACT co-creation trajectories.

**Figure 3 ijerph-19-07150-f003:**
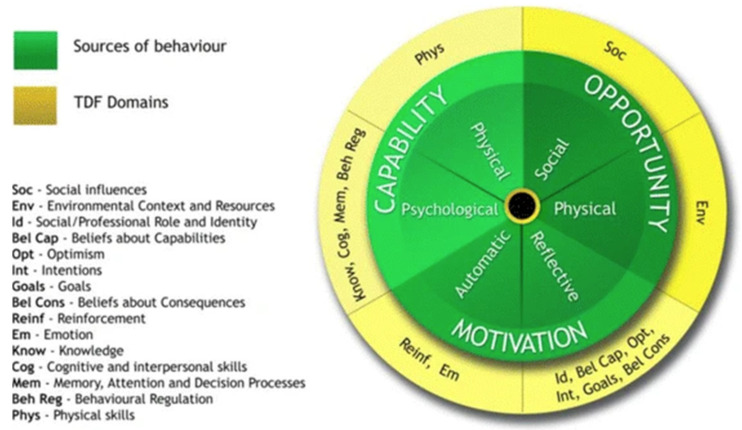
The three sources of behaviour and the TDF Domains of the Behaviour Change Wheel. “Adapted with permission from [47]. 2017, Lou Atkins”.

**Figure 4 ijerph-19-07150-f004:**
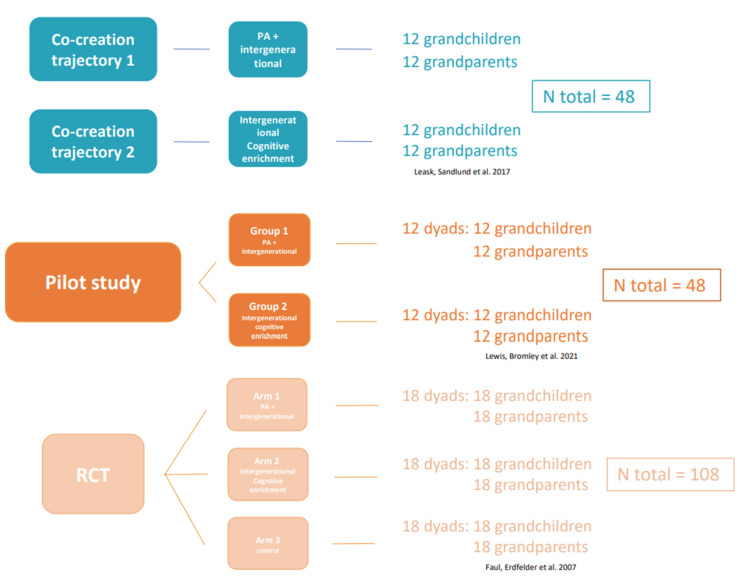
Overview of the study design and number of participants [58,59,60].

**Figure 5 ijerph-19-07150-f005:**
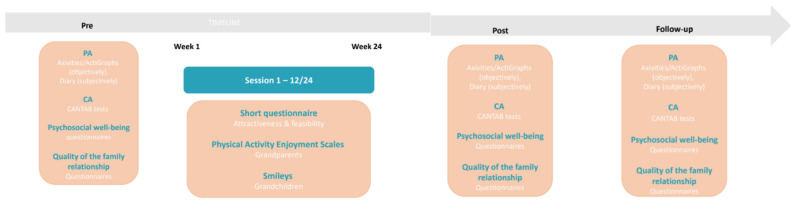
Overview of the RCT with the pre- (baseline), post- (after 24 weeks) and follow-up (after 36 weeks) measurements.

## Data Availability

Not applicable.

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
