# Peer review of "The GRANDPACT Project: The Development and Evaluation of an Intergenerational Program for Grandchildren and Their Grandparents to Stimulate Physical Activity and Cognitive Function Using Co-Creation"

_ijerph, 2022, doi:10.3390/ijerph19127150_

Round 1

Reviewer 1 Report

Congratulation for the very clear outline of your study. I wish you successful completion of your study. I missed, however, details on the registration of the trial, and strongly recommend a trial registration after all the details of the randomized controlled trials are clarified.

I have some minor remarks, only.

p2, line 68: Please clarify MVPA.

p2, line 93 et seq. and p3, line 102 et seq.: Could you please shortly comment on the level of evidence of these studies. Are these randomized controlled trials?

Figure 2: Well done!

p9, line 370: Are 10.000 steps daily appropriate/realistic in older people?

p11, line 499: Are accelerometers used only in the sessions?

p13, line 550: I have doubts that an effect size of 0.15 is clinically meaningful. Do you have any reference for this? Could you also translate your effect size in an absolute score (like minutes of physical activity)?

p17, line 762 et seq.: I recommend using the TIDieR checklist for describing the intervention program. This could help then finally presenting/publishing the results of your study.

Good luck!

Reviewer 2 Report

The reviewed article has potential cognitive value, and its subject matter corresponds to the profile of the Journal. The Authors presents in great detail the process of developing a program aimed at increasing the physical and cognitive activity of children and the elderly. The added value of this program is the simultaneous inclusion in the proposed forms of activity of people representing two generations: grandchildren and grandparents. The Authors justify in detail the need to develop the program and ground it in a specific theory (which is important for its evaluation and the construction of similar solutions).

The Authors' intention was not to show the results of the program, or to describe it, but to present a protocol illustrating the individual steps of its creation. I understand that in the next article they will present the results of the individual studies, whose organization they describe in this work. I wonder if it wouldn’t be more beneficial to leave a description of the program construction procedure here and add more details about the program itself, and in another work to describe the analyses aimed at checking the usefulness of the program. Reading, I had the impression that I would learn about the research results including the recruitment of the respondents.

Please check the order of scoring used in the article and explain abbreviations, eg.: MVPA, MOEV, GRAY.

Graph 5 should appear before the description of co-creation (or it should be mentioned in the description of co-creation). The data in the graph is relevant for all three studies. Likewise, ethical consent applies to all three studies, which should be clearly stated in the article.

I do not know why the authors present the headline Results, which does not include the research results but only the research intention. How will the Authors (statistically) check the mediating role of co-PA (p. 16, 683-686)?

In the discussion, the Authors write about the weaknesses and strengths of their research, but without knowing their results, we cannot comment on them.

The part devoted to research is clearly weaker: it needs to be organized and possibly moved to a separate article.

Reviewer 3 Report

Figure 3 exposes the Behavior Change Wheels, which is very interesting. But the authors should analyze some other factors that have a great impact on human behavior, such as technology, which was widely used in the pandemic context. A new model will result.

In lines 276-277 the authors specify 2-3 weeks break in between sessions, which I consider too long and the results can not be eloquent for this research.

Starting from section 2.3.1. the author should introduce a new section - 3 Methodology.

This article is purely theoretical, which is interesting, but we need to find out the results of the PA  impact on children and grandparents.  Another research might be done based on a survey as the authors suggest.

All figures must have a higher resolution. The content can not be read.

We suggest that the authors resubmit the paper after testing this program at least on a pilot group, otherwise this article represents an only a literature review  

Round 2

Reviewer 3 Report

Congratulations for your research!